# Apolipoprotein D as a Potential Biomarker in Neuropsychiatric Disorders

**DOI:** 10.3390/ijms242115631

**Published:** 2023-10-26

**Authors:** Eva del Valle, Nuria Rubio-Sardón, Carlota Menéndez-Pérez, Eva Martínez-Pinilla, Ana Navarro

**Affiliations:** 1Department of Morphology and Cell Biology, University of Oviedo, 33006 Oviedo, Spain; valleeva@uniovi.es (E.d.V.); nuria199510@hotmail.com (N.R.-S.); menendezperezcarlota@gmail.com (C.M.-P.); anavarro@uniovi.es (A.N.); 2Instituto de Neurociencias del Principado de Asturias (INEUROPA), 33006 Oviedo, Spain; 3Instituto de Investigación Sanitaria del Principado de Asturias (ISPA), 33006 Oviedo, Spain

**Keywords:** bipolar disorders, lipocalins, mayor depression disorders, serum, schizophrenia

## Abstract

Neuropsychiatric disorders (NDs) are a diverse group of pathologies, including schizophrenia or bipolar disorders, that directly affect the mental and physical health of those who suffer from them, with an incidence that is increasing worldwide. Most NDs result from a complex interaction of multiple genes and environmental factors such as stress or traumatic events, including the recent Coronavirus Disease (COVID-19) pandemic. In addition to diverse clinical presentations, these diseases are heterogeneous in their pathogenesis, brain regions affected, and clinical symptoms, making diagnosis difficult. Therefore, finding new biomarkers is essential for the detection, prognosis, response prediction, and development of new treatments for NDs. Among the most promising candidates is the apolipoprotein D (Apo D), a component of lipoproteins implicated in lipid metabolism. Evidence suggests an increase in Apo D expression in association with aging and in the presence of neuropathological processes. As a part of the cellular neuroprotective defense machinery against oxidative stress and inflammation, changes in Apo D levels have been demonstrated in neuropsychiatric conditions like schizophrenia (SZ) or bipolar disorders (BPD), not only in some brain areas but in corporal fluids, i.e., blood or serum of patients. What is not clear is whether variation in Apo D quantity could be used as an indicator to detect NDs and their progression. This review aims to provide an updated view of the clinical potential of Apo D as a possible biomarker for NDs.

## 1. Introduction

In the past decades, there has been growing interest in the study of neuropsychiatric disorders (NDs) since they have become the first cause of disability and the second cause of death worldwide [1]. A European meta-analysis published in 2011 estimated that 38.2% of the population suffers from at least one brain disorder each year, i.e., mood or developmental, which means around 164 million people [2,3,4]. The most frequent disorder is anxiety (14%), followed by insomnia (7%), major depression (6.9%), somatoform disorders (6.3%), and alcohol and drug dependence (4%); however, attention deficit hyperactivity disorder (ADHD) stands out in the youngsters (5%) [2]. Most recent studies, based on data from the World Health Organization (WHO), place the incidence of this group of pathologies at almost 50%, an increase that would be related to increasingly frequent natural disasters or conflict-induced humanitarian crises in numerous countries, as well as the recent Coronavirus Disease (COVID-19) pandemic, among many other factors [5,6,7].

From a clinical point of view, there are several circumstances that could be behind the neuropsychiatric symptoms, including metabolic alterations, the consumption of poisons/toxins, and general medical conditions such as neoplasm, trauma or infections [8]. This complexity constitutes a handicap for the diagnosis and treatment of NDs. Primary information provided from gene expression or brain activity imaging techniques is useful but with limitations. Thus, there is a current effort focusing on the discovery of potential biomarkers for improving prevention, diagnosis, drug response, and drug development for NDs. In this sense, finding biomarkers for NDs would help to predict outcomes, stratify groups of patients, and make decisions regarding treatments and therapies. Undoubtedly, an ideal biomarker is an intracellular or extracellular molecule that shows measurable differences between physiological and pathological states. Protein profiling in serum, plasma, urine, saliva, or cerebrospinal fluid (CSF) in NDs is a promising field of research [4]. In addition, not only proteomics but all the “omics”, including genomics, transcriptomics, metabolomics, and epigenetic techniques, are powerful tools in the search of biomarkers [9]. The big question is whether there is any molecule that meets all these requirements and can be used in clinical practice.

The apolipoprotein (Apo) family is a specialized group of proteins that associates with lipids and mediates several steps in lipid metabolism. So far, there are 22 known members of the family but only Apo E [10], Apo J [11], and Apo D [12] are expressed at high levels in the nervous system [13]. Several studies indicate that of candidate psychiatric biomarkers detected using proteomic techniques, cholesterol and associated proteins, specifically apolipoproteins, may be of interest. Cholesterol is necessary for brain development and its synthesis continues at a lower rate in the adult brain. Specifically, Apo D is the component of lipoproteins responsible for lipid transport [14,15], whose implication in NDs has gained interest in recent years [16,17,18,19,20,21]. In this review, we will focus on what is known so far about the relationship between Apo D and NDs, and which are the features of this protein that could make it feasible to be used as a new biomarker for these pathologies.

## 2. Apolipoprotein D

### 2.1. Basic Information

Apo D is an “atypical” apolipoprotein since its molecular structure revealed that it belongs to the lipocalin superfamily [12,22]. Human Apo D was first isolated and partially characterized by McConathy and Alaupovic in 1973 from plasma HDLs [22] and subsequently by Albers et al. in 1981 [23], but is widely expressed throughout the human body, as reported by Drayna et al. in 1986 [12]. These authors found Apo D mRNA in the brain, liver, kidney, intestine, pancreas, placenta, spleen, and adrenal and lachrymal glands. Apart from plasma, Apo D has been detected in different human fluids such as tears [24], CSF [25], perilymph [26], sweat [27], urine [28], breast gross cyst fluid [29], and in cerumen [30]. Sites of N-glycosylation were found at Asn45 and Asn78. Interestingly, Apo D glycosylation patterns are tissue specific, with a molecular weight ranging from 29 to 32 kDa depending on glycosylation level. This feature is important because while 29 kDa Apo D is synthesized in the central nervous system (CNS), the 32 kDa form is found in the CSF and plasma. Therefore, detection of 29 kDa Apo D in human fluids would be a direct consequence of its leakage across an injured blood–brain barrier (BBB), making Apo D a perfect biomarker for BBB damage [31].

At the molecular level, Apo D shows the lipocalin characteristic ligand binding structure, which classifies it as a kernel lipoprotein able to bind arachidonic acid (AA), cholesterol, and steroids. No specific ligand has been identified, but Apo D binds with high affinity to AA, progesterone, and retinoic acid, whereas with reduced affinity to pregnenolone and some eicosanoids [32,33,34]. Moreover, Apo D also forms a complex with LCAT (Lecithin Cholesterol Acyl Transferase), the main enzyme responsible for plasma cholesterol esterification. This association allows Apo D to regulate AA metabolism, preventing its transformation in cholesterol esters [35]. Although Apo D is usually found as a monomeric protein, it has also been described in dimeric and tetrameric forms [36,37]. In fact, the Apo D dimers seem to be related to the antioxidant function of this lipocalin, protecting cells from lipid hydroperoxides [36]. Since Apo D could exert different roles depending on its location, it has been defined as a multi-ligand multi-function protein [21].

Apo D expression is regulated by various factors as its promoter region contains several response elements (estrogens, glucocorticoids, progesterone, vitamin A, and vitamin D). In vitro experiments have shown that the gene is under the control of serum-responsive elements during cellular growth arrest [38]. Apo D is also modulated by Apo E3 and Apo E4 isoforms, but not Apo E2. It has been observed that isoforms E3 and E4 inhibited Apo D promoter activity in normal and stress conditions. An inverse correlation between Apo D and Apo E mRNA expression during development and in the cortex, hippocampus, plexus choroid, and cerebellum of the mouse brain was also demonstrated [39]. Recently, a new promoter region in the Apo D gene that, according to authors, would work in tandem with a 5′UTR has been discovered in the nervous system in response to oxidative stress [40].

### 2.2. Apo D in Nervous System

The mammalian brain is one of the organs of major expression for Apo D, but this lipocalin is also expressed in the peripheral nervous system during regeneration and remyelination [41]. As a lipid carrier, Apo D could have an important role in lipid transport during axonal regeneration. In fact, Apo D, together with A-IV, A-I, and E, accumulates in the sciatic nerve of rats during the regenerative period following a lesion as a part of a crucial complex for the mobilization and metabolic regulation of lipids [42,43,44]. Ganfornina et al., in 2010, studying sciatic nerves of Apo D-KO mice, demonstrated that it is necessary for the correct maintenance of myelin sheaths and for the functional integrity of axons along aging [41]. These authors also confirmed that Apo D is upregulated after an injury of the nerve. At this point, Apo D would contribute to the recovery of the axonal function, favoring the lipid exchange during degenerative and regenerative phases and controlling the magnitude and duration of the inflammatory response in the damaged area [41,44].

In the human CNS, Apo D is located in perivascular cells, astrocytes, oligodendrocytes, and some neurons in both physiological and pathological conditions [45,46]. Of note, Apo D has been identified as the most upregulated gene in humans, mice, rats, and rhesus macaques along the aging brain process [47,48] and, more importantly, Apo D levels are increased under pathological situations, i.e., after kainate damage [49] or traumatic brain injury [50].

### 2.3. Apo D in Neurodegenerative Diseases

During the last decades, numerous in vitro [51,52,53,54,55] and in vivo [21,56,57,58,59,60] studies in different neuropathologies such as Alzheimer’s disease (AD), Parkinson’s disease (PD), or multiple sclerosis (MS) have postulated that Apo D would be part of the cellular defense machinery against oxidative stress and inflammation, with a clear neuroprotective function [21,61,62,63]. For example, Apo D is upregulated in the CSF, hippocampus, and cerebral cortex of AD patients in correlation with the Braak degeneration stage [64]. Interestingly, increased Apo D expression was found in oligodendrocytes and astrocytes, and also a colocalization with amyloid plaques, probably in an attempt to protect neurons against β-amyloid-induced cytotoxicity [36,65]. In the case of PD, the glial cells surrounding dopaminergic neurons of the substantia nigra show an increased Apo D immunosignal [58]. In addition, changes in Apo D levels were described in both the plasma and brain of a murine model of Niemann–Pick Type C disease [21].

Analyses of CSF from patients with chronic inflammatory demyelinating polyneuropathy and Guillain–Barré Syndrome revealed an increase in Apo D content [66]. Similar data, as well as an intrathecal increment of Apo D, were found in MS patients [66,67]. Intriguingly, Apo D indices (calculated to exclude any influence of blood–CSF barrier leakage) measured in CSF correlate with MS duration but not with disability or age [66]. The increase in Apo D presence could respond to a protective mechanism, being the highest at the time of the first clinical exacerbation. Another example is the study of Do Carmo et al. (2008), whose results in a mouse model of acute encephalitis by coronavirus infection showed that Apo D is upregulated in the brain of these animals, returning to normal levels when the virus is cleared [68].

Finally, Apo D has also been associated with other neurodegenerative lesions, as it has been shown in different experimental models of stroke, entorhinal cortex lesions, or maternal hyper and hypothyroidism [21,60].

## 3. Role of Apo D as a Biomarker in Neuropsychiatric Disorders

Current evidence strongly suggests the increase of Apo D expression in NDs such as Schizophrenia (SZ), Bipolar Disorders (BPD), Major Depressive Disorders (MDD), or Autism Spectrum Disorder (ASD), a group of diseases that shares some cardinal pathological features including oxidative stress, excitotoxicity, myelin dysfunction, cholesterol imbalance, and apoptosis, and where it seems to play a fundamental role as a neuroprotective protein. The existing lack of biomarkers for the effective and early detection of NDs makes it necessary to search for new predictive molecules, and, in this sense, Apo D could be a good candidate. An updated view of the role of Apo D as a potential biomarker for different NDs, summarized in Figure 1, will be treated in the following sections.

### 3.1. Apo D in Schizophrenia

SZ is a severe and long-term brain condition characterized by changes in how a person thinks, perceives, and interprets reality that affects approximately 20 million individuals globally [69]. Major symptoms that may appear continuously or in relapsing episodes include psychosis, hallucinations, delusions, disorganized speech, lack of motivation, or cognitive deficits [70]. Researchers speculate that a variety of genetic and environmental factors contribute to the development of SZ [71]. While there is no cure for SZ, ongoing research is paving the way for innovative and more secure treatment options. However, the success of these treatments involves, without a doubt, adequate diagnostic methods. Nowadays, clinical diagnosis predominantly relies on subjective evidence, encompassing self-reported experiences and observed behavioral abnormalities, followed by psychiatric evaluations [72]. Consequently, there is a pressing need for objective and specific diagnostic tests based on biomarkers to enhance the accuracy of SZ diagnosis in clinical practice [69]. Although the pathological mechanisms of the disease are not fully understood, multiple lines of evidence indicate that disruptions in membrane lipids and their metabolism caused by excessive oxidative stress likely play a role in the development of SZ, affecting neurotransmission and complex brain function [70,73].

Taking this into consideration, Thomas et al., in 2001, described, for the first time, a selective increase of Apo D quantity in some brain areas of patients with SZ [74]. They took samples from the left hemisphere of 20 subjects diagnosed with SZ and analyzed Apo D levels using Western blot (WB) and ELISA techniques. They found significant increases in Apo D amount in the prefrontal cortex (BA9 and BA46; 46% increase), orbitofrontal cortex (BA11; 23.9%), amygdala (42.8%), thalamus (31.7%), and caudate, precisely the brain areas most affected by the pathophysiological changes related with SZ [75]. Remarkably, Apo D serum levels were lower in these patients than in control ones, suggesting that the local Apo D increment may act as a kind of compensatory response against systemic insufficiencies in lipid metabolism. However, it is important to keep in mind that most of the patients in these studies were medicated with neuroleptic drugs, and a gradual accumulation in both Apo D mRNA and protein was reported in response to chronic clozapine (CLZ) administration in the brains of rodents [76]. In humans, a significant increment of Apo D serum levels was found in never-medicated patients when compared with controls and even higher in those chronic patients treated with the atypical antipsychotic drug CLZ versus drug-free ones [77]. These findings seem to support the view that Apo D could be acting as an antioxidant neuroprotective molecule to prevent AA peroxidation and, at the same time, as a biological marker in an indirect way [62].

Furthermore, Apo D high levels have also been found in a study that looked for biomarkers of psychosis risk in blood. In fact, Apo D has been identified, together with matrix metalloproteinase 7, as the only two molecules of 117 (related to hormonal responses, inflammation, growth, oxidative stress, and metabolism) that showed a significant difference between people with clinical high-risk symptoms who developed psychosis and those who, despite presenting symptoms, did not do so [78]. A meta-analysis for exploring potential biomarkers of SZ in human peripheral fluids confirmed Apo D as one of the 20 proteins that were altered in the serum and plasma of SZ patients compared with controls. Despite these results, authors proposed several proteins as possible biomarkers for SZ, but Apo D was not among them [79]. Relatedly, Raiszadeh et al., in 2012, performed a proteomic analysis of sweat in SZ subjects [80]; Apo D was one of the five proteins that sweat and serum had in common. However, expression levels of Apo D in sweat samples were not high enough to take into consideration the analysis of this lipocalin as an SZ biomarker. These results are quite surprising as Apo D has been reported by other authors, after dermicidin and Apo J, as the third most abundant protein in human sweat, representing 15% of the total of secreted proteins [81].

The differences in the results of the mentioned studies could be explained by different factors such as the analysis methodologies, the number of subjects, the phase of the disease, or the medication. Once again, data about pharmacological treatments of patients is interesting since it has been shown that some drugs, like haloperidol (HAL), reduce Apo D expression in the hippocampus, piriform cortex, and caudate-putamen, while others, such as risperidone (RISP) or olanzapine (OLZ), produce the opposite effects in rat brain [82]. In fact, the negative effects of HAL over cell morphology, as well as Apo D quantities, could be restored with RISP or OLZ post-treatment. Even more, pre-treatments with these atypical antipsychotic drugs also prevent Apo D reduction caused by HAL but not in such an effective way as post-treatments. It is known that CLZ, RISP, and OLZ are better choices than HAL for SZ treatment due to the negative effects on cognition and extra-pyramidal symptoms caused by the last one. Therefore, Apo D increase could be involved in the molecular mechanisms that underlie the positive effects of CLZ, RISP, and OLZ in SZ treatment, e.g., to sequester free AA in the cell, preventing its release and its entry into inflammatory pathways such as the cyclooxygenase [18,35,82,83]. Anyway, it would be interesting to know the effect of more drugs on Apo D expression to determine if this lipocalin could be a good biomarker for SZ.

In the CSF, Apo D has been identified as a potential biomarker for cerebral amyloid angiopathy [84], chronic pain [85], MS [86], and idiopathic normal pressure hydrocephalus [87], among other brain-related pathologies. When CSF was analyzed in the search for SZ markers, using iTRAQ-based proteomic and mass spectrometry techniques, Apo D was identified as one of the ten proteins that showed differential expression across the clinical dopaminergic spectrum but did not show linearity or statistical significance to be considered as a biomarker probably due to the low number of patients included in these experiments [88].

While it is true that due to the invasiveness of the techniques, using Apo D as a biomarker for diagnosing SZ in brain tissue or CSF does not provide an advantage over other biomarkers, the ability to detect changes in Apo D levels with just a blood sample and a protein profiling assay places it in an advantageous position compared to other molecular markers whose detection involves methodologically complex and expensive assays, such as RNA sequencing.

### 3.2. Apo D in Bipolar Disorders

BPDs are a group of chronic mental health conditions that cause abnormal shifts in mood, ranging from extreme highs (mania) to extreme lows (depression) episodes [71]. BPDs affect the quality of life of millions of adults worldwide, negatively impacting day-to-day social interactions and potentially leading to premature mortality from cardiovascular disease or suicide [89]. The exact cause of these pathologies is unknown, but genetics play a key role, accounting for 70% of the risk for BPD as heritable [90]. Regarding BPD physiopathology, recent studies suggest that immune-inflammatory changes induce structural brain alterations at the limbic network that compromise dopamine and serotonin neurotransmitter signaling, which would explain the symptoms and the cognitive deterioration of patients [90,91].

The study of the Apo D brain levels in post-mortem samples of BPD subjects, in the same areas as the authors did previously for SZ, demonstrated differences between the two NDs [75]. In the case of BPD, an increase of Apo D expression was observed in the dorsolateral prefrontal (BA9; 273% increment), lateral prefrontal (BA46; 111% increment), and in the parietal cortices (BA40; 123% increment) of BPD patients vs. healthy controls. In contrast, no significant increase was observed in the orbitofrontal (BA11; 37.9% increment) and cingulate cortices (BA24; 57.7% increment), and no changes were appreciated in the amygdala or the thalamus [74,75].

In regard to the potential of Apo D as a serum biomarker for BPD, results found in the literature are quite contradictory. Dean and co-workers, in 2008, using ELISA immunological assays, did not observe differences in Apo D levels between BPD vs. SZ patients or their controls [92]. Interestingly, medications did not seem to have an effect on the amount of Apo D in the plasma of BPD subjects. Contrarily, Knöchel et al., in 2017, performed a nanoliquid chromatography–multiple reactions monitoring mass spectrometry (nano-LC–MRM-MS) and observed that BPD patients exhibited higher Apo D levels in serum than those diagnosed with SZ (*p* < 0.050) [93]. Similar results have been reported in a recent study; Apo D is found elevated in the serum of BPD patients compared to controls [94]. In this case, the proteins were separated by gel electrophoresis and then analyzed by mass spectrometry. At this point, it would be interesting to know the type of BPD of the patients and the number of subjects included in these studies, as well as to explore a little further the real influence of methodologies on the results obtained.

Notwithstanding, it is important to remember that Apo D conformation is not the same in the post-mortem brains and in plasma, and, consequently, their behavior may be completely different. Thus, a systematic review and a meta-analysis on MS-based proteomics applied to human peripheral fluids to assess potential biomarkers of BPD showed that Apo D is present in plasma and serum, and it was among the 258 proteins differentially expressed in blood-related samples (plasma, serum, and PBMCs) when studying BPD patients vs. controls [79]. Finally, and as far as we know, Apo D has not been studied in other human fluids from BPD subjects.

### 3.3. Apo D in Major Depressive Disorder

MDD or clinical depression is a ND characterized by a persistent low mood, feelings of sadness, and loss of interest that condition the daily functioning and quality of life of patients [95]. It is considered the most common mental illness worldwide; the number of cases has increased by almost 50% over the past 30 years [96]. Since MDD is very variable in the lifetime course and recurrences are frequent, the early diagnosis of the initial depressive episode is instrumental in prescribing individualized treatments and preventing multiple episodes [95]. Until today, all attempts carried out to identify an applicable biomarker panel to the disease have failed [97].

The MDD pathophysiology includes dysfunction of the hypothalamic–pituitary–adrenal (HPA) axis, neurotransmitter metabolism disorder, oxidative stress, and neuroinflammation [98], processes in which Apo D could participate thanks to its function as a neuroprotective protein. However, there is no data about Apo D levels in brain tissue from patients with MDD since all studies have been done in serum samples. In this sense, Apo D was first reported in relation to MDD by Xu et al. in 2012, when these authors observed a 1.69-fold change increase in Apo D levels in MDD first-episode never-medicated patients compared with healthy controls using iTRAQ 2D LC-MS/MS. Nonetheless, when the proteins were studied by WB, the differences were not significant [99]. In a similar study (first-onset drug-naïve MDD patients) using multiplexed immunoassay profiling and LC-MS(E), despite Apo D being present above the limits of detection in all samples studied, no significant changes in Apo D levels were found in MDD patients vs. controls [100]. Surprisingly, in 2016, Apo D was identified as one of the six proteins that could be used as serum biomarkers in MDD, with a 68% accuracy, according to the authors [101]. In this study, blood samples of 50 non-medicated MDD patients and their corresponding controls were analyzed by LC–MS/MS to obtain a serum proteome profiling. With this method, a group of differentially expressed proteins was found, most of them implicated in inflammatory response and lipid transport, Apo D among them [79,99]. Moreover, the authors tested, with a logistic regression model, the robustness of these proteins (Apo B, Apo D, CP, GC, HRNR, and PFN1) in terms of accuracy, sensitivity, and specificity, obtaining values of 68, 67, and 69%, respectively [79]. Overall, the study not only detects a group of proteins that could be used as biomarkers of MDD but also implies that the modulation of inflammatory and immune systems, as well as lipids metabolism, are implicated in the pathophysiology of MDD. In this way, high levels of Apo D in the serum of non-medicated depressed patients may reflect a neuroprotective response to oxidative stress, systemic inflammation, neurovascular dysfunction, and BBB permeability [102]. It is important to bear in mind that certain treatments for depressive disorders have effects on lipid metabolism, so it would be interesting to know what happens with Apo D in MDD-medicated patients [103,104].

The study of Lee et al., performed by liquid chromatography–tandem mass spectrometry and label-free quantification in plasma from BPD and MDD subjects, took into account the effects of medication over protein expression (antipsychotics, mood stabilizers, antidepressants, or benzodiazepines) and demonstrated that Apo D expression was not affected for any of the antipsychotic drugs. However, it was negatively associated with the total scores of the Hamilton depression rating scale (HAM-D) and with the anhedonia/retardation and guilt/agitation scores of this scale [105]. In fact, Apo D expression seems to decrease with the worsening of symptoms, which would make it impossible to exert its neuroprotective role.

Clinical evidence suggests that some people suffering from MDD attempt suicide. It has been reported that alterations in cholesterol levels could play a crucial role in suicidal behavior, as shown by the significant association between (i) low levels of peripheral circulating cholesterol in patients with suicidal and violent conduct [106,107], and (ii) low levels of central cortical cholesterol in violent suicide completers [108]. There is little data about Apo D and suicidal behavior. Genome-wide expression profiling using DNA from samples of the prefrontal cortex (BA47) of ten suicidal subjects and eight matching controls showed that six genes were downregulated in the suicides, including Apo D (−1.57-fold change). Unfortunately, Apo D data were not validated by real-time PCR, as happened with Apo E (−1.63-fold change) or sortilin 1 (1.86-fold change). Anyway, Apo D and Apo E low levels could be an indicator of failure in cholesterol transport, as expected in this type of disorder [109].

In the last decade, there has been an increasing interest in the use of secreted membrane vesicles, collectively called extracellular vesicles (EVs), as potential diagnostic tools for NDs [110,111,112]. Although the content of these EVs is cell-specific, it is mainly represented by proteins and nucleic acids, but also by lipids and metabolites (for a review, see [113]). Recently, the possible role of EVs in MDD has been the subject of study. In fact, EVs are packed with RNA (mainly miRNAs) and proteins related to altered processes of MDD, i.e., energy metabolism, neuro-inflammation, neurogenesis, and the maintenance of BBB, Apo D among them [114]. In physiological conditions, EVs secreted by astrocytes transport Apo D to neurons to enhance neuroprotection under oxidative stress [115]. Since EVs can successfully pass the BBB, Li et al. (2023) propose that they may be considered as transport carriers of Apo D and other neuroprotective factors in precise brain-related treatments [114]. Therefore, it would be a priority to study in depth the presence of Apo D in these vesicles in MDD and to investigate the possibility that this protein could act as a biomarker of the pathology.

### 3.4. Apo D and Other Neuropsychiatric Disorders

In view of the potential role of Apo D as a biomarker for SZ, BPD, or MDD, some laboratories are focusing on this protein but in other NDs, namely ASD, anxiety, and addictions.

#### 3.4.1. Apo D in Autism Spectrum Disorder

ASD is a complex neurodevelopment condition that can manifest with deficits in social communication, restricted interest, and repetitive behaviors. The estimated prevalence of ASD has been increasing in the past two decades, mainly affecting children under 3 years of age, although symptoms may improve over the adult’s life [116]. The multifactorial etiology, which includes genetic and non-genetic factors, together with the heterogeneity of its symptoms, complicates early diagnosis and the choice of the most effective and appropriate interventions [117]. Unfortunately, today, there are no biological markers of ASD that are objectively measurable and easily quantifiable to avoid the delay in diagnosis [118].

Lipid peroxidation, abnormalities in prostaglandin metabolic pathways, oxidative stress, and alterations in the formation and maintenance of synaptic structures have been associated with the pathogenesis of ASD, so Apo D could play a role in this pathology due to its ability to interact with AA, as already mentioned in this review [119]. On the one hand, the study of Esnafoglu and Cırrık (2022) demonstrated no significant differences in Apo D serum levels between ASD patients and controls when these were measured using ELISA [120]. The authors pointed out that one of the limitations of the study was the small sample (39 individuals). On the other hand, they also did not find differences in the analysis of the presence of Apo D in some brain tissues of ASD subjects compared to controls. This fact may be attributed to the age of the subjects, ranging from 3 to 13 years old, which makes it more difficult to observe significant variations. First, it has been shown that Apo D expression is age-dependent [121]. Second, Apo D serum level in the controls was 1.39 ± 0.18 µg/mL, while in healthy adults, it was between 5 and 23 mg/100 mL [122].

#### 3.4.2. Apo D in Anxiety Disorders

The origin of some anxiety disorders could be traced as early as the three first weeks after birth. In mammals, enhanced postnatal care during this period operates by suppressing microglial inflammation, and is associated with reduced anxiety [123]. Mice lacking inflammation-suppressing factor IRF2BP2 in microglia (KO) and the wild-type (wt) ones, subjected to enhanced postnatal care, show differential expression of Apo D [123]. Thus, its expression is lower in the hypothalamus of wt than of KO mice, which would prove the implication of Apo D not only in neurodevelopment, as we referred to above, but its contribution to a minor anxiety state. The nucleus accumbens is a brain area studied in relation to anxiety disorders involved in motor function but also in emotion and reward. A mouse model lacking type 5 adenylyl cyclase (AC5) in the nucleus accumbens presents an anxiolytic phenotype. In these mice, Apo D is the second most downregulated gene after cholecystokinin [124]. Once again, a reduction in anxiety levels in an animal model of disease is accompanied by a downregulation of the Apo D gene, which would make it unequivocally a downward marker of the pathology. As far as we know, there are no specific studies on humans about Apo D and anxiety disorders, but the work of Huang et al. gives us some indirect information. These authors analyzed the lipoprotein concentration of patients with anxiety states and found a lower HDL concentration in affected women vs. their respective controls [125]. Since Apo D is a component of HDL, it would be interesting to study this fact in depth.

#### 3.4.3. Apo D in Addictions

Critically, there are noticeable variations between studies examining Apo D levels in patients with addictions to different substances, i.e., drugs of abuse, cigarettes, or alcohol. As we mentioned previously, Apo D levels could be modulated by some medical drugs [76,77,82]. Moreover, some results suggest that astrocytes could be a target for drugs of abuse and casually that this cell type is one of those with the highest expression of Apo D. All this leads to the question of whether this apolipoprotein is also affected by drugs of abuse.

Most of the studies in this field have centered on alcohol and smoking effects. Variations in Apo D gene levels were first reported in 2000 in a study with tissues from the superior prefrontal cortex of alcoholic subjects using four different microarrays [126]. Notwithstanding, the Apo D gene was downregulated in all assays; authors were not sure that the changes observed would be only due to alcoholism and not to the aging process or to other confounding variables [127]. The downregulation of the Apo D gene in the frontal cortex of alcoholics was confirmed by Flatscher-Baden et al. in 2005 [128]. Furthermore, as alcoholism and smoking usually coexist, there are studies about the influence of smoking on the expression of genes modified by alcohol consumption. For instance, Apo D mRNA expression increased in the prefrontal cortex of smokers in a significant way, independently of alcohol ingestion, when compared with the non-smoker alcoholic group [129,130].

Regarding other substances of abuse, references in the literature are scarce, but it has been observed that morphine also upregulates the Apo D gene in the lateral hypothalamus of wt mice vs. those KO for mu opioid receptor after a chronic treatment [131]. Finally, Apo D is also expressed in the hippocampus of cocaine users, although the changes were not significant when compared with controls [132].

## 4. Final Remarks

In the CNS, Apo D acts not only as a lipid carrier (the typical role of a lipocalin) but also seems to be a neuroprotective molecule in oxidative stress situations, as well as necessary for the correct myelination process. It is clear that Apo D levels (protein or gene) are altered in several NDs (Table 1). Of note, if variations in Apo D amounts in serum from SZ and MDD patients are confirmed, Apo D could be used as a biomarker for the diagnosis of these two NDs. However, the studies of Apo D’s usefulness as a biological marker in other NDs are few and not standardized, an essential requirement to create a panel of biomarkers for diagnosis and discrimination between NDs. Obviously, the serum is the ideal media for the analysis of Apo D levels, so larger cohorts are needed for the studies. It should be remembered that Apo D glycosylation patterns are tissue-specific, which makes it a perfect candidate as a serum biomarker of BBB damage. In this sense, it is also important to classify subjects according to the type, stage, or grade of the disorder due to the possibility that Apo D could act as a “first response” protein with higher levels at the onset of the pathology. As some drugs (medical drugs and substances of abuse) influence Apo D expression, it is a priority to discriminate between drug-naïve and medicated patients, and also know the type of medication that the individuals are receiving and for how long. Furthermore, since the Apo D gene has estrogen response elements in its promoter region, sex must be taken into account when analyzing the results. Finally, to avoid inconsistencies between studies, authors should pay special attention to the chosen methodology. The state-of-the-art mass spectrometry methods and RNA arrays are very useful for selecting candidate biomarker molecules, but the data should always be checked by real-time PCR.

Future perspectives in this field of research necessarily involve more studies about the presence of Apo D in human fluids in different NDs to establish an expression pattern that could be used as a biomarker to help in the diagnosis of these pathologies but also to consider Apo D as a neuroprotective molecule that could help in the treatment of NDs due to its antioxidant and anti-inflammatory properties.

## Figures and Tables

**Figure 1 ijms-24-15631-f001:**
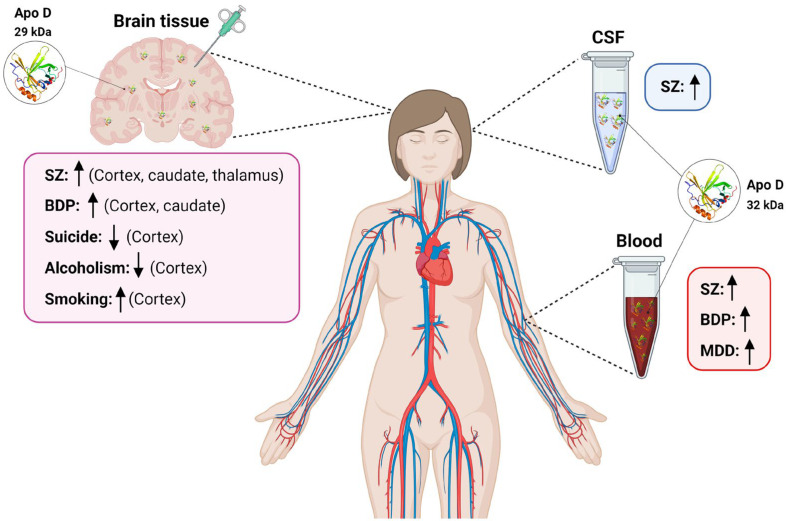
Scheme summarizing variations in Apo D levels in blood, CSF, and human brain tissues found in different neuropsychiatric diseases. SZ: schizophrenia; BPD: Bipolar disorders; CSF: cerebrospinal fluid; MDD: Major depressive disorder. ↑: increased Apo D levels; ↓: decreased Apo D levels. Created with BioRender.com.

**Table 1 ijms-24-15631-t001:** Apo D variations in neuropsychiatric disorders.

	Brain Tissue		Serum/Plasma		CSF		Other	
SZ	↑ BA9 and Caudate; BA46, BA11 and Thalamus (human)	[74,75]	↓ (mouse)	[76]	ns (human)	[66]	= Sweat (human)	[81]
	= Substantia nigra, BA18, CA1, CA3, Subiculum, Parahyppocampal gyrus and Cerebellum; BA40 and BA24 (human)	[74,75]	↑(human)	[76]				
BPD	↑ BA9 and Caudate; BA46 and BA40 (human)	[74,75]	↑ (human)	[93,94]	nd		nd	
	= BA18; Thalamus (human)	[74,75]	= (human)	[92]				
	Ns BA11 and BA24 (human)	[75]						
MDD	nd		↑ (human)	[101]	nd		nd	
			ns (human)	[99,100]				
Suicide	↓ BA10 ^(^*^)^ (human)	[109]	nd		nd		nd	
ASD	nd		nd		nd		↑ Amniotic fluid ^(^*^)^ (human)	[119]
Alcoholism	↓ Superior frontal cortex ^(^*^)^ (human)	[126]	nd		nd		nd	
	↑↓ Frontal cortex ^(^*^)^ (human)	[127]						
	↓ Frontal cortex ^(^*^)^ (human)	[128]						
	↑↓ Motor cortex ^(^*^)^ (human)	[127]						
Smoking	↑ Prefrontal cortex ^(^*^)^ (human)	[129]						
	↓ Hippocampus ^(^*^)^ (mouse)	[130]						
Morphine addiction	↑ Lateral hypothalamus ^(^*^)^ (mouse)	[131]						
Cocaine addiction	ns Hippocampus ^(^*^)^ (human)	[132]						

^(^*^)^: gene (otherwise refers to protein); **↑**: increases; **↓**: decreases; =: does not change; nd: not determined; ns: not significant; SZ: schizophrenia; BPD: Bipolar disorders; MDD: Major depressive disorder; BA: Brodmann’s area; ASD: Autism Spectrum Disorders.

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
