# Peer review of "Apolipoprotein D as a Potential Biomarker in Neuropsychiatric Disorders"

_ijms, 2023, doi:10.3390/ijms242115631_

Round 1

Reviewer 1 Report

Comments and Suggestions for Authors

1. The author mentioned thatApo D is a perfect biomarker for BBB damageIt is confusing,whether the Apo D can pass the BBB? Otherwise, the tested Apo D in body fluid caused by   BBB leak.

2. In my view,the part 2. Apolipoprotein Dcan be separate as three part:1. Basic information;2. Apo D in nervous system; 3. Apo D in neurodegenerative disease.

3. As the known makers, more details about ApoD in in neurodegenerative disease should be provided.

3. To provide a better understanding,please give the right citation of Table 1 and Figure 1.

4. In 3.4, it still need some subtitles.

5. The author mentioned that “ Apo D gene has estrogen response elements in its promoter region”. Is there any studied about the sex-biased function of ApoD in neuropsychiatric disorders?

6. The abstract need a better explanation.

7. Line 334:The author mentioned that “  EVs can successfully pass the BBB. Whether Apo D have been tested in exosomes.

8. As small RNAs can pass the BBB,the author should provide the description about ”  miRNAs-related ApoD pathways in neuropsychiatric disorders diagnosis”.

9. In my view, the author should added the basic molecular construction, and different protein weight in Figure 1. Again, the figure1 should be moved to the part2.

10. The table should use the three line types.

11. Compare to other markers, what‘s the advantages and disadvantages in the application of ApoD in SD diagnosis? The author should explained more clear.

Comments on the Quality of English Language

none

Author Response

Point by point response to reviewers' comments:

The author mentioned that “Apo D is a perfect biomarker for BBB damage” It is confusing whether the Apo D can pass the BBB? Otherwise, the tested Apo D in body fluid caused by BBB leak.

Response: we acknowledge this comment. The detection of 29kDa Apo D in human fluids would be a direct consequence of its leakage across a damaged BBB. The sentence has been rewritten in the revised version of the manuscript in order to avoid any misunderstandings.

In my view the part “2. Apolipoprotein D” can be separate as three part:1. Basic informationï¼›2. Apo D in nervous system; 3. Apo D in neurodegenerative disease.

Response: we appreciate the suggestion that has been taken into account and section 2 has been separated into three subsections in the revised version of the manuscript.

As the known makers, more details about Apo D in neurodegenerative disease should be provided.

Response: thanks for raisin this issue that has been considered (briefly and with new references) in the revised version of the manuscript.

To provide a better understanding, please give the right citation of Table 1 and Figure 1.

Response: thanks for the comment that has been considered in the revised version of the manuscript.

In 3.4, it still need some subtitles.

Response: we appreciate the suggestion that has been taken into account in the revised version of the manuscript.

The author mentioned that “Apo D gene has estrogen response elements in its promoter region”. Is there any studied about the sex-biased function of Apo D in neuropsychiatric disorders?

Response: thanks for raisin this issue. There are some studies about Apo D in which the subjects were separated by sex, but those studies are about neurodegenerative diseases not neuropsychiatric conditions.

The abstract need a better explanation.

Response: we appreciate this suggestion. As per reviewer’s request the abstract has been rewritten in the revised version of the manuscript.

Line 334:The author mentioned that “EVs can successfully pass the BBB”. Whether Apo D have been tested in exosomes.

Response: thanks for the comment, unfortunately there are almost no studies that test Apo D in exosomes. In fact, as far as we know the presence of Apo D has only been detected in EV produced by astrocytes, as we mentioned in the manuscript, and there is no information about its ability to pass through the BBB. For a review see Nassar A, Kodi T, Satarker S, Chowdari Gurram P, Upadhya D, Sm F, Mudgal J, Nampoothiri M. Astrocytic MicroRNAs and Transcription Factors in Alzheimer's Disease and Therapeutic Interventions. Cells. 2022 Dec 17;11(24):4111. doi: 10.3390/cells11244111. PMID: 36552875; PMCID: PMC9776935.

As small RNAs can pass the BBB the author should provide the description about “miRNAs-related Apo D pathways in neuropsychiatric disorders diagnosis”.

Response: we appreciate the comment but “miRNAs-related Apo D pathways” constitutes a huge and complex field of study yet to be investigated, even more in the neuropsychiatric disorders diagnosis context. Honestly, the information that we can add to the manuscript would be merely speculative, and we think that it would not contribute anything to the scope of this work.

In my view, the author should have added the basic molecular construction, and different protein weight in Figure 1. Again, the figure1 should be moved to the part2.

Response: thanks for the suggestion, we have added this information to the Figure 1 in the revised version of the manuscript, but since Part 3 is more closely related to the function of Apo D as a biomarker of neuropsychiatric diseases, which is the topic of Figure 1, we honestly think that 1 it would fit better in the first paragraph of Part 3, as long as the reviewer agrees"

The table should use the three line types.

Response: we appreciate the suggestion that has been taken into account in the revised version of the manuscript.

Compare to other markers, what ‘s the advantages and disadvantages in the application of Apo D in SD diagnosis? The author should have explained more clear.

Response: we acknowledge this comment; this issue has been explained in more detail in the revised version of the manuscript.

Reviewer 2 Report

Comments and Suggestions for Authors

This is well written and comprehensive review. Only small comments: 1) 199: it unusual to use word "shocking" in scientific papers, use e.g. "suprising"; 2) line 243: i think Authors meant her "no significant" but not "no significate", please change; 3) Table: i dont understand why in some rows Authors write arrow and ns, e.g. SCH, CSF, increasing arrow ns (human), so did it increase or no? Please check whole table because in few places there are this kind of discrepancies and they are really confusing.

Author Response

Point by point response to reviewers' comments:

This is well written and comprehensive review. Only small comments:

Thanks for the positive comments

1) 199: it unusual to use word "shocking" in scientific papers, use e.g. "suprising"

Response: we appreciate the suggestion that has been taken into account in the revised version of the manuscript.

2) line 243: i think Authors meant her "no significant" but not "no significate", please change

Response: thanks for detect this error that has been fixed in the revised version of the manuscript.

3)Table: i dont understand why in some rows Authors write arrow and ns, e.g. SCH, CSF, increasing arrow ns (human), so did it increase or no? Please check whole table because in few places there are this kind of discrepancies and they are really confusing.

Response: thanks for raisin this issue that has been considered in the table of the revised version of the manuscript to avoid any misunderstandings.

Round 2

Reviewer 1 Report

Comments and Suggestions for Authors

The author have answered all the comments